ecology/ecosystems

fisheries-induced, evolution, population dynamic, ecosystem functioning, community composition, intraspecific variability

**Author for correspondence:**
Charlotte Evangelista
e-mail: charlotte.evangelista0@gmail.com

†These authors jointly supervised this work.

# Ecological ramifications of adaptation to size-selective mortality

Charlotte Evangelista[1], Julia Dupeu[2],
Joakim Sandkjenn[1], Beatriz Diaz Pauli[1,3],
Anders Herland[1], Jacques Meriguet[4,5],
Leif Asbjørn Vøllestad[1,†] and Eric Edeline[2,6,†]

[1]Centre for Ecological and Evolutionary Synthesis (CEES), Department of Biosciences, University of Oslo, Oslo, Norway
[2]Sorbonne Université, CNRS, INRAE, IRD, Université Paris Est Créteil, Institut d'Ecologie et des Sciences de l'Environnement de Paris (iEES-Paris), Paris, France
[3]Department of Biological Science, University of Bergen, Bergen, Norway
[4]CEREEP Ecotron Île-de-France, UMS CNRS/ENS, Saint-Pierre-lès-Nemours, France
[5]Institut de Biologie de l'Ecole Normale Supérieure, CNRS, INSERM, PSL Research University, Paris, France
[6]ESE, Ecology and Ecosystem Health, INRAE, Agrocampus-Ouest, Rennes, France

CE, 0000-0002-9586-0868

Size-selective mortality due to harvesting is a threat to numerous exploited species, but how it affects the ecosystem remains largely unexplored. Here, we used a pond mesocosm experiment to assess how evolutionary responses to opposite size-selective mortality interacted with the environment (fish density and light intensity used as a proxy of resource availability) to modulate fish populations, prey community composition and ecosystem functions. We used medaka (*Oryzias latipes*) previously selected over 10 generations for small size (harvest-like selection; small-breeder line) or large size (large-breeder line), which displayed slow somatic growth and early maturity or fast somatic growth and late maturity, respectively. Large-breeder medaka produced more juveniles, which seemed to grow faster than small-breeder ones but only under high fish density. Additionally, large-breeder medaka had an increased impact on some benthic prey, suggesting expanded diet breadth and/or enhanced foraging abilities. As a consequence, increased light stimulated benthic algae biomass only in presence of large-breeder medaka, which were presumably better at controlling benthic grazers. Aggregated effect sizes at the community and ecosystem levels revealed that the ecological effects of medaka evolution were of similar magnitude to those induced by the environment and fish introduction. These findings indicate the important environmental dependency of evolutionary response to opposite size-selective mortality on higher levels of biological organizations.

# 1. Introduction

Harvesting by hunters or fishers is one of the most pervasive anthropogenic disturbance to wild populations [1,2]. Removal of the largest individuals often selects in parallel for slower somatic growth and earlier maturation [3–6]. Although the consequences of size-selective harvesting for population and community dynamics have been discussed [7], the potential impacts of harvest-induced evolution *per se* on the whole ecosystem remain surprisingly overlooked [8]. This topic is important because intraspecific variation in phenotypic traits can have stronger ecological implications than the removal or the reduction of the species itself [9–12]. However, the contribution of evolution to this intraspecific biodiversity-ecosystem functioning (iBEF) relationships is seldom quantified (but see [13]).

Consumers can regulate ecosystem dynamics through top-down, consumptive and/or bottom-up, nutrient-mediated effects. The strength of these effects is modulated by consumer body size and somatic growth rate [14,15], precisely those traits that evolve in response to harvesting. Hence, understanding the ecosystem-level consequences of adaptation to size-selective mortality requires studying the concomitant consumptive- and nutrient-mediated effects of consumers. Large-bodied and fast-growing consumers usually intensify top-down effects due to their high feeding rates [9,16,17], and can also modulate the structure of resource communities by foraging on a greater variety of prey (i.e. larger diet breadth) [18,19]. In parallel with their stronger top-down effects, large-sized consumers excrete nutrients at higher *per capita* rates than small-sized ones due to allometric scaling of metabolism [15] and this consumer-induced nutrient cycling can boost primary production [20]. Hence, harvest-induced evolution towards slower somatic growth rates and smaller body size has the potential to weaken both the top-down and bottom-up effects that harvested species induce to their environment.

Harvest-induced evolution may also alter the ability of the harvested population to cope with changes in their environment. For instance, harvesting removes individuals from the population, leading to reduced intraspecific competition and increased resource availability for survivors. Therefore, most fisheries management models assume that harvesting increases biomass production by populations [21]. However, harvest-induced evolution towards slower somatic growth might decrease the ability of exploited populations to cope with competition and to increase production in response to lower density and higher food availability [6,22,23]. To date, however, there is scant knowledge on the context-dependency of evolution-induced effects of harvesting (Environment × Evolution; but see [24]).

Here, we assessed whether and how evolutionary responses to size-selective mortality and the environment (i.e. population density and primary production) interact to modulate fish production and excretion, prey community composition and ecosystem functioning. To do so, we performed a three-month pond mesocosm experiment using medaka (*Oryzias latipes*), a small Asian omnivorous fish species. We used two lines of medaka originating from a size-selection experiment performed over 10 generations. The selection procedure consisted of mimicking either fishing mortality where small-bodied individuals are favoured to reproduce (small-breeder SB line), or a more natural mortality regime that favours large-bodied individuals (large-breeder LB line) [25,26]. Under controlled laboratory conditions, the LB and SB lines evolved different life-history traits and behaviours: small-breeder medaka grew slower, matured earlier and were less willing to forage than the large-breeder medaka [23,25,27]. We hypothesized that:

(1) Medaka biomass production in terms of somatic growth and recruitment (larvae and juvenile recruitment) would be lower in populations composed of SB medaka (adapted to small-sized selection). This is because earlier reproduction at a smaller body size is associated with reduced fecundity and often reduced larval viability [23,28]. Further, a lower willingness to forage in the SB medaka may result in less energy available for somatic growth and reproduction.
(2) SB populations would excrete nutrients at lower rates than LB ones because they are composed of individuals with smaller body size [15].
(3) The effect of size selection on fish biomass production and excretion rates would depend on environmental conditions. This is because SB medaka have lower capacity to cope with competition and lower foraging ability than LB medaka [23,27]. Therefore, we expected biomass production to be more negatively impacted by high fish density in the SB than in the LB line. We further expected SB medaka to benefit less from increased primary production (and increased availability of primary consumers higher up in the food chain). Finally, we expected that line-by-environment effects would influence population excretion rates, notably because these latter are expected to increase with an increasing number of fish [29].

(4) The direct and interactive effects of harvest-induced evolution and the environment (i.e. fish density and primary production) on fish biomass production and excretion would propagate through the ecosystem, resulting in SB populations having less impacts on prey community and ecosystem processes. In particular, we expected SB population to have reduced top-down and bottom-up (i.e. consumer-nutrient mediated) effects due to their lower willingness to forage [27] and their lower role in nutrient recycling (hypothesis 2).

# 2. Material and methods

## 2.1. Size-dependent selection and fish rearing

The experimental medaka originated from two lines artificially size-selected over 10 generations under controlled laboratory conditions to ensure that differences between lines were genetically rather than environmentally induced (temperature: 26°C, photoperiod: 14 h light/10 h dark, density: 14–17 fish per 3 l tank, feeding: ad libitum with a mixed diet of dry food and living *Artemia salina* and/or *Turbatrix aceti*). The selection procedure consisted in removing the largest or the smallest breeders, hence producing two lines with distinct life-history strategies: the small-breeder line (resulting in slower growth rate and earlier maturation) where only small-bodied individuals were allowed to reproduce, and the large-breeder lines (resulting in faster growth and delayed maturation). Specifically, at 60 days post-hatching (dph), among a total of at least 20 families per line, the 10 families with the largest (large-breeder line) or smallest (small-breeder line) average standard body length (SL) were kept. At 75 dph, individuals within each of the selected families were measured and the largest-bodied (large-breeder line) or the smallest-bodied (small-breeder line) mature males ($n = 2$ or 3) and females ($n = 2$ or 3) were used as breeders for the next generation (further details available in [25]). On average at 75 dph, SL was 20.7 mm in small breeders and 22.0 mm in large breeders (a 5.7% difference), and the probability of being mature was 91.7% in small breeders and 77% in large breeders (a 18.0% difference) [25].

On 27 June 2017, fish from the 11th generation (dubbed F11) were checked for maturity based on the development of secondary sexual characters [30]. For each line, 180 mature fish (initial standard length: mean ± s.d.; $SL_i$ in small-breeder = 18.9 mm ± 1.4; $SL_i$ in large-breeder = 19.4 mm ± 1.4; ANOVA: $F_{1, 358}$ = 13.70, $p < 0.001$) were selected to generate 24 populations composed of individuals from the same line (48 populations in total), but from distinct families to limit inbreeding (mean kinship coefficient = 0.23 ± 0.1 and 0.17 ± 0.1 s.e.m. in LB and SB populations, respectively; further details available in [26]). Fish were anaesthetized with MS-222 and marked using visible implant elastomer (VIE; Northwest Marine Technology, Shaw Island, WA, USA) to render each fish individually identifiable within each population and to allow the calculation of individual somatic growth rate. Fish from the same population were pooled in a 3 l tank and maintained at the laboratory until the beginning of the experiment when they were released into an outdoor mesocosm (electronic supplementary material, figure S1.A).

## 2.2. Outdoor mesocosm experiment

The outdoor experiment was conducted at the CEREEP-Ecotron Ile de France (Saint-Pierre-les-Nemours, France; cereep.bio.ens.psl.eu) using 60 circular mesocosms (500 l, 0.8 m deep, 1.0 m diameter). Medaka line (small-breeder SB and large-breeder LB), fish density (high HD, low LD) and light intensity (high light HL and low light LL) were manipulated in a factorial design, with six replicates of each of the eight treatment combinations (figure 1). Additionally, 12 fishless mesocosms (6 HL and 6 LL) were set up as controls to quantify the effects of fish introduction and light intensity conditions on the ecosystem. Due to space constraints, treatments were arranged in five blocks (i.e. 12 mesocosms per block), within which two treatments were replicated twice. High- and low-density treatments consisted of 12 and 3 fish per mesocosm (or 3.2 mg fish l$^{-1}$ ± 0.3 s.d. and 0.9 mg fish l$^{-1}$ ± 0.1 s.d.), respectively, with an equivalent sex ratio (HD: 8 females and 4 males; LD: 2 females and 1 male). Light supply was used to modulate primary production while avoiding too high growth of filamentous algae. Light was manipulated using shade nets with different mesh size that allowed the passage of 92% (high light intensity HL) and 70% of ambient light (low light intensity LL).

All mesocosms were filled simultaneously from 4 to 6 April 2017 with tap water (100 l) and oligotrophic water from a local pond (300 l). The pond water was pre-filtered through 150 µm mesh to remove large invertebrates, zooplankton and debris. Mesocosms were also supplied with 2 l of a mature sediment mixture including benthic invertebrates and 2 l of a homogenized and concentrated

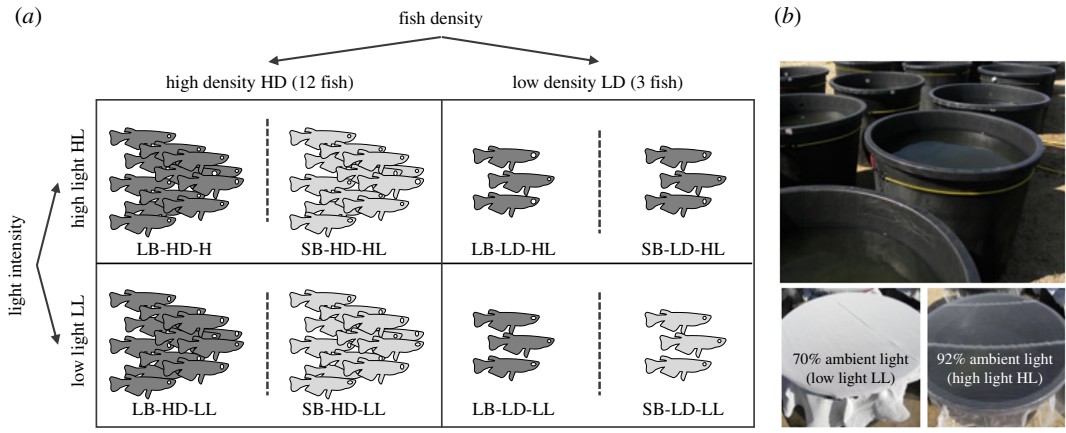

**Figure 1.** (a) Design of the mesocosm experiment used to test the effects of Line × Density and Line × Light intensity on fish characteristics and ecological variables. (b) Pictures of the outdoor mesocosms (upper picture) and shade nets used to manipulate light intensity (lower pictures). Note that fishless mesocosms ($n = 12$, 6 in low light and 6 in high light) are not shown here.

mixture of zooplankton (Copepoda and Cladocera) collected from local ponds. In each mesocosm, two floating shelters made of wool threads (30 cm length) provided spawning substrate and two floating brushes made of plastic threads provided shade and shelter (protection) for larvae. Each mesocosm was then covered with a net (see details above) and given three months to mature before fish were introduced. On 12 June, all mesocosms were enriched with 2 ml of a liquid mixture of 0.32 µg P l$^{-1}$ as KH$_2$PO$_4$ and 6.32 µg N l$^{-1}$ as NaNO$_3$ to favour primary production.

During the experiment, we observed that the light intensity treatment (manipulated using shade nets) modulated water evaporation because mesocosms with LL intensity had a higher water volume than the HL mesocosms ($F_{1,54} = 824.36$, adj $p > 0.001$; electronic supplementary material, table S1.A and figure S1.B). Therefore, where required ($n = 36$ mesocosms), we adjusted the water level in these mesocosms to *ca* 300 l by adding osmosis water (added volume: mean = 15.6 l ± 6.3 s.d.). Water was added after sampling of the various ecosystem metrics. In so doing, we maintained an adequate habitat for fish and limited the effect of the water addition on the nutrient and chlorophyll-a concentrations. Importantly, water temperature remained similar between light intensity treatments ($F_{1,54} < 0.01$, adj $p = 0.986$; electronic supplementary material, table S1.A and figure S1.B).

## 2.3. Ecosystem and prey community metrics

In each mesocosm, whole-mesocosm metabolism (daily community respiration CR24 and gross primary productivity GPP), benthic and pelagic algae biomasses and nutrient concentrations in the water column (ammonium NH$_4^+$ and soluble reactive phosphorus SRP) were quantified every two weeks until the end of the experiment, starting one week after fish introduction (table 1; further details available in electronic supplementary material, S1). Due to logistic constraints, whole-mesocosm metabolism (CR24 and GPP) was not estimated during the third sampling event, while nutrient concentration measurements were not available for the first and third sampling events (electronic supplementary material, figure S1.A).

All zooplankton and zoobenthos individuals collected at the end of the experiment were used for the calculation of total abundances, but abundances were also calculated by aggregating individuals into the following taxonomic groups: Copepoda and Cladocera (zooplankton), Ostracoda, Chironomidae larvae, Nematoda, Hydrachnidia, Planorbidae, Corbiculidae and Ephemeroptera (zoobenthos). The diversity of zoobenthos was calculated using Simpson's diversity index bounded between 0 (low diversity) and 1 (high diversity) (table 1; further details available in electronic supplementary material, S1). Two zooplankton samples (one from the SB-HD-HL and one from the SB-LD-HL treatment) were lost.

## 2.4. Population excretion rates

At the end of the experiment (three-month duration), all fish (marked fish and new unmarked fish born during the experiment) were removed with hand nets from the mesocosms. We then immediately quantified population NH$_4^+$ and SRP excretion rates by placing all the fish collected from a mesocosm

**Table 1.** General description of the different variables collected during the experiment and of the main linear and generalized mixed-effects models (LMMs and GLMMs) used to test the interactions between Line × Density and Line × Light intensity on these variables.

| | variables | general methodology | no. of sampling events ($n$) | statistical analyses models; *distributions* | structure of the random effects |
|---|---|---|---|---|---|
| fish characteristics | adult survival | visual count of adults | 5 ($n = 720$) | GLMM *binomial (log link)* | (1\|sampling/block/mesocosm) |
| | recruitments | visual count of larvae and juveniles | 5 ($n = 720$) | GLMMs *Poisson (log link)* | (1\|sampling/block/mesocosm) |
| | growth rate | recaptured fish | 1 ($n = 330$) | LMM *Gaussian* | (1\|block/mesocosm) |
| | | offspring (final length) | 1 ($n = 289$) | LMM *Gaussian* | (1\|block/mesocosm) |
| | excretion rates[a] ($NH_4^+$ and SRP) | colorimetric analyses of excretion trials [31] | 1 ($n = 48$) | LMMs *Gaussian* | (1\|block) |
| prey community metrics | zooplankton taxa | taxonomic identification | 1 ($n = 46$) | GLMMs *Poisson or negative binomial (log link)* | (1\|block) |
| | zoobenthos taxa | taxonomic identification | 1 ($n = 48$) | GLMMs *Poisson or negative binomial (log link)* | (1\|block) |
| | zoobenthos diversity | Simpson's diversity [32] (bounded between 0 and 1) | 1 ($n = 48$) | GLMM *Beta distribution (logit link)* | (1\|block) |
| ecosystem metrics | metabolism (GPP and CR) | diel change in dissolved oxygen [33] | 5 ($n = 240$) | LMMs *Gaussian* | (1\|sampling) + (1\|block/mesocosm) |
| | algae biomasses[a] (benthic and pelagic) | *in situ* chlorophyll-a concentrations | 6 ($n = 288$) | LMMs *Gaussian* | (1\|sampling) + (1\|block/mesocosm) |
| | nutrient concentrations[a] ($NH_4^+$ and SRP) | colorimetric analyses of water samples | 4 ($n = 192$) | LMMs *Gaussian* | (1\|sampling) + (1\|block/mesocosm) |

[a]$\log_{10}$-transformed data.

in a plastic bag filled with 500 ml of spring bottled water. Plastic bags were closed and immediately immersed in their respective mesocosm to keep temperature constant and limit stress due to visual contact. After 40 min of incubation, water samples were filtered (Whatman GF/F filter) from the bags and were analysed for $NH_4^+$ and SRP concentrations following the protocol described in electronic supplementary material, S1. For each population, population excretion rates of $NH_4^+$ and SRP ($\mu g\ h^{-1}$) were calculated following Vanni *et al.* [31]:

$$\text{Excr}_x = \frac{([X]_{\text{pop}} - [X]_{\text{control}}) \times V}{t},$$ (2.1)

where $[X]_{\text{pop}}$ and $[X]_{\text{control}}$ are the concentrations ($\mu g\ l^{-1}$) of the element $X$ quantified for fish population and control (a similar bag filled with only spring bottled water, i.e. without fish), respectively. $V$ is the volume (l) of spring bottled water in the plastic bag and $t$ is the time of incubation (h). For each block, one control bag was used to assess the background concentration of $NH_4^+$ and SRP.

## 2.5. Fish biomass production

After the excretion trials, fish were measured for final standard length ($SL_f \pm 1$ mm), checked for marks, categorized as recaptured or offspring (i.e. including larvae, juvenile and new adult) and euthanized with an overdose of MS-222. The somatic growth rate (mm month$^{-1}$) of each recaptured fish ($n = 330$; the average survival rate of 92%) was calculated as follows:

$$\text{growth rate} = \frac{SL_f - SL_i}{t},$$ (2.2)

where $SL_f$ and $SL_i$ are the final and initial standard length and $t$ is the duration of the experiment (three months).

During the experiment, the number of fish within each mesocosm was quantified from visual counts on five occasions, repeated three times on each occasion at the morning (around 9.00), noon (around 12.00) and mid-afternoon (16.00). Specifically, every two weeks (starting two weeks after fish were introduced; electronic supplementary material, figure S1.A), the same operator counted the number of larvae (SL < 10 mm), the number of juveniles (10 < SL < 15 mm) and the number of adults (SL > 15 mm) over 2 days (blocks A, B and C the first day, and blocks D and F the second day). The size limit among the three stages corresponded to visually different morphologies that characterized each category [30]. The time spent for observing each mesocosm was reduced to 3 min to standardize the sampling effort.

## 2.6. Statistical analyses

Statistical analyses were performed using R v. 4.0.3 [34]. The effects of size-selected line, fish density and light intensity on fish characteristics (i.e. fish biomass production and population excretion rates), ecosystem metrics and prey community composition (i.e. invertebrate abundances and Simpson's diversity index) were tested using a combination of linear and generalized mixed-effects models (LMMs and GLMMs, respectively). The models were fitted with different random structures depending on whether the data were collected on a single event or on multiple occasions during the experiment (table 1). In all models, we included the two-way interactions Line × Density and Line × Light intensity. We did not provide any hypotheses for higher-order interactions and thus they were not included. The model with somatic growth rate of recapture fish as response variable also included 'initial fish length' as a fixed effect, while the number of fish (centred to zero mean) used in the excretion trials was added as a fixed effect in the models with population excretion rates.

Linear mixed-effects models (LMMs) were fitted using the 'lme4' package (v. 1.1.25; [35]), and GLMMs using the 'glmmTMB' package (v. 1.0.2.1; [36]). The significance of all factors was evaluated using 'Anova' from the 'car' package (v. 3.0.10; [37]). Specifically, Wald chi-square ($\chi^2$) or F Type III tests were performed when interactions were significant, while Type II tests were performed when interactions were not significant [38]. All interactions were maintained in the final models, independently of whether they were significant or not. Significant interactions were further analysed using pairwise comparisons carried out using 'emmeans' from the 'emmeans' package (v. 1.5.2.1; [39]). Given the large number of comparisons involved, the false discovery rate procedure [40] was applied to correct for alpha inflation using the 'p.adjust' function (base-package v. 4.0.4). Assumption of linearity and homogeneity of variance on residuals from LMMs were checked visually, and $\log_{10}$ transformations were applied where

required (table 1). QQ-plots were also used to detect outliers and three data points (one for daily community respiration, one for Cladocera abundance and one for Nematoda abundance) were removed from the analyses, with no qualitative impact on the results (electronic supplementary material, S4). Diagnostics for models fitted with 'glmmTMB' were performed using the 'DHARMa' package (v. 0.3.3.0; [41]) and for each variable, the best distribution (e.g. generalized Poisson or negative binomial distributions) was chosen based on likelihood-ratio tests. Plots display raw data and predicted values (ŷ) from the models were computed using the 'predict' function (base-package v. 4.0.4). Additional LMMs and GLMMs were used to test the effects of fish introduction (two-level factor: fish absence versus fish presence), and light intensity in fishless mesocosms on ecosystem metrics and prey community composition (electronic supplementary material, S2).

Hedge's effect sizes were calculated to compare the magnitude of the effects of each treatment (i.e. size selection, fish density and light intensity) and fish introduction on ecosystem metrics (i.e. ecosystem metabolism, algae biomasses and nutrient concentrations) and prey community composition (i.e. invertebrate abundances and Simpson's diversity index). A standardized effect size was thus computed for each response variable and each treatment (with two levels each) using the following formula [42]:

$$\text{Hedges' } g = \frac{m_2 - m_1}{\sqrt{(n_2 - 1)\text{SD}_2^2 + (n_1 - 1)\text{SD}_1^2/(n_2 + n_2 - 1)}}, \tag{2.3}$$

where $m$ is the group mean and SD is the group standard deviation of the response variable for each level of one treatment. An absolute mean effect size was calculated for all ecosystem or invertebrate variables by averaging the effect sizes of all response variables. The absolute mean effect size of each treatment was interpreted as negligible if $|g| < 0.20$, small if $|g| = 0.20$–$0.30$, medium if $|g| = 0.30$–$0.80$ and large if $|g| \geq 0.80$ [10,12]. The differences between the absolute effect sizes of each treatment were tested using paired $t$-tests.

# 3. Results

## 3.1. Fish biomass production and excretion

Juvenile abundance was significantly affected by the Line × Density and Line × Light intensity interactions (table 2). Specifically, juvenile abundance was higher in LB than in SB populations, but only in the high-density or high-light intensity treatments (approx. 3- and 15-fold differences, respectively; figure 2a,b; electronic supplementary material, table S3.A). These results indicate that juvenile abundance in SB populations was more negatively impacted by increasing density than that of LB population (figure 2a), and also did not benefit from increasing light intensity (figure 2b). The Line × Density interaction had a marginally significant effect on offspring length (figure 2c; table 2), which seemed to be longer in LB than in SB populations, but only under high medaka density (adj $p = 0.053$; electronic supplementary material, table S3.A). Analyses of the number of larvae indicated that eggs from the two lines probably started to hatch at the same time during the experiment (no significant Line × Day effect; table 2). Therefore, density-dependent offspring length variations between the two lines indicate that LB offspring seemed to grow faster than SB ones in the high-density treatment. Size selection (alone or in interaction) had no effect on adult survival probability, larvae abundance, somatic growth of recaptured fish and population excretion rates (table 2).

## 3.2. Prey community and ecosystem metrics

At the community level, the abundance of Nematoda and Ostracoda prey were approximately 2.5- and 3-fold significantly higher in SB than in LB populations, suggesting higher predation rates from the LB populations (figure 3a,b; table 2). The total abundance of zoobenthos was affected by the Line × Light intensity interaction (table 2). Specifically, in the low-light treatment, zoobenthos abundance was higher in SB than in LB populations (figure 2c; electronic supplementary material, table S3.A). None of the other community metrics were affected by size selection (table 2).

At the ecosystem level, there was only a significant effect of the Line × Light intensity interaction on the biomass of benthic algae (table 2). Benthic algae biomass was higher in mesocosms with LB medaka than those with SB medaka, but only in the high-light treatment (figure 3d; electronic supplementary material, table S3.A).

**Table 2.** Analysis-of-variance table derived from the models used to assess the effects of the size-selected line (large-breeder LB and small-breeder SB), fish density (high HD and low LD) and light intensity (high HL and low LL) on fish characteristics, prey community composition and ecosystem metrics. Significant $p$-values adjusted for the false discovery rate (adj $p$) are in italics. *Indicates $log_{10}$ transformed response variables, † indicates generalized Poisson distribution; ‡ indicates binomial distribution; § indicates negative binomial distribution; # indicates beta distribution.

**fish characteristics**

LMMs/GLMMs with generalized Poisson† or binomial distribution‡

| predictors | adult survival probability‡ | | | number of larvae† | | | no. juveniles† | | |
|---|---|---|---|---|---|---|---|---|---|
| | $\chi^2_{d.f.}$ | p | adj p | $\chi^2_{d.f.}$ | p | adj p | $\chi^2_{d.f.}$ | p | adj p |
| intercept | — | — | — | — | — | — | $4.03_1$ | 0.045 | 0.145 |
| Line | $1.64_1$ | 0.201 | 0.458 | $0.45_1$ | 0.503 | 0.737 | $41.05_1$ | <0.001 | *<0.001* |
| Density | $168.81_1$ | <0.001 | *<0.001* | $21.79_1$ | <0.001 | *<0.001* | $7.98_1$ | 0.005 | *0.022* |
| Light intensity | $1.52_1$ | 0.217 | 0.490 | $0.23_1$ | 0.635 | 0.831 | $3.20_1$ | 0.074 | 0.216 |
| Day (scale) | — | — | — | $1034.39_1$ | <0.001 | *<0.001* | — | — | — |
| Line × Density | $0.36_1$ | 0.549 | 0.772 | $5.05_1$ | 0.025 | 0.088 | $27.21_1$ | <0.001 | *<0.001* |
| Line × Light intensity | $0.29_1$ | 0.592 | 0.812 | $0.43_1$ | 0.514 | 0.741 | $9.83_1$ | 0.002 | *0.010* |
| Line × Day (scale) | — | — | — | $0.15_1$ | 0.702 | 0.875 | — | — | — |

| predictors | somatic growth rate (recaptured fish) | | | final length (offspring) | | |
|---|---|---|---|---|---|---|
| | $F_{d.f.,\ d.f.res}$ | p | adj p | $F_{d.f.,\ d.f.res}$ | p | adj p |
| intercept | — | — | — | — | — | — |
| initial length | $60.55_{1,316.14}$ | <0.001 | *<0.001* | — | — | — |
| Line | $0.78_{1,33.18}$ | 0.385 | 0.670 | $1.44_{1,16.78}$ | 0.246 | 0.524 |
| Density | $235.82_{1,69.89}$ | <0.001 | *<0.001* | $1.00_{1,23.72}$ | 0.328 | 0.615 |
| Light intensity | $1.21_{1,30.67}$ | 0.281 | 0.568 | $1.60_{1,19.40}$ | 0.222 | 0.493 |
| Line × Density | $0.98_{1,71.24}$ | 0.326 | 0.615 | $7.22_{1,21.11}$ | 0.014 | 0.053 |
| Line × Light intensity | $0.58_{1,31.80}$ | 0.451 | 0.702 | $0.05_{1,20.08}$ | 0.822 | 0.922 |

(Continued.)

**Table 2.** (*Continued.*)

| predictors | population NH$_4^+$ excretion* | | | population SRP excretion* | | |
|---|---|---|---|---|---|---|
| | $F_{\text{d.f., d.f.res}}$ | $p$ | adj $p$ | $F_{\text{d.f., d.f.res}}$ | $p$ | adj $p$ |
| intercept | — | — | — | — | — | — |
| fish number (centred to zero mean) | $7.46_{1,36.09}$ | 0.010 | 0.646 | $0.88_{1,36.74}$ | 0.355 | 0.646 |
| Line | $1.16_{1,36.22}$ | 0.288 | 0.569 | $0.70_{1,37.62}$ | 0.409 | 0.674 |
| Density | $11.34_{1,36.20}$ | 0.002 | *0.010* | $1.01_{1,37.49}$ | 0.322 | 0.615 |
| Light intensity | $0.47_{1,36.04}$ | 0.498 | 0.737 | $0.08_{1,36.23}$ | 0.780 | 0.897 |
| Line × Density | $0.86_{1,36.35}$ | 0.360 | 0.633 | $0.04_{1,38.47}$ | 0.845 | 0.930 |
| Line × Light intensity | $4.93_{1,36.22}$ | 0.033 | 0.111 | $0.93_{1,37.65}$ | 0.342 | 0.633 |

prey community metrics

GLMMs with generalized Poisson† or negative binomial§ or beta distribution#

| predictors | total zooplankton§ | | | Copepoda† | | | Cladocera† | | |
|---|---|---|---|---|---|---|---|---|---|
| | $\chi^2_{\text{d.f.}}$ | $p$ | adj $p$ | $\chi^2_{\text{d.f.}}$ | $p$ | adj $p$ | $\chi^2_{\text{d.f.}}$ | $p$ | adj $p$ |
| intercept | — | — | — | — | — | — | — | — | — |
| Line | $0.06_1$ | 0.814 | 0.922 | $0.03_1$ | 0.856 | 0.936 | $2.22_1$ | 0.136 | 0.362 |
| Density | $0.16_1$ | 0.690 | 0.874 | $0.05_1$ | 0.818 | 0.922 | $0.53_1$ | 0.468 | 0.710 |
| Light intensity | $2.07_1$ | 0.150 | 0.372 | $3.30_1$ | 0.069 | 0.206 | $0.66_1$ | 0.417 | 0.674 |
| Line × Density | $0.09_1$ | 0.764 | 0.889 | $0.66_1$ | 0.417 | 0.674 | $0.01_1$ | 0.942 | 0.964 |
| Line × Light intensity | $3.65_1$ | 0.056 | 0.170 | $2.09_1$ | 0.148 | 0.372 | $0.11_1$ | 0.746 | 0.889 |

| predictors | total zoobenthos§ | | | Ostracoda† | | | Chironomidae§ | | |
|---|---|---|---|---|---|---|---|---|---|
| | $\chi^2_{\text{d.f.}}$ | $p$ | adj $p$ | $\chi^2_{\text{d.f.}}$ | $p$ | adj. $p$ | $\chi^2_{\text{d.f.}}$ | $p$ | adj $p$ |
| intercept | $118.44_1$ | <0.001 | <0.001 | — | — | — | — | — | — |
| Line | $7.93_1$ | 0.005 | 0.022 | $8.65_1$ | 0.003 | *0.017* | $0.02_1$ | 0.875 | 0.945 |
| Density | $0.40_1$ | 0.527 | 0.754 | $1.04_1$ | 0.309 | 0.604 | $2.18_1$ | 0.140 | 0.362 |
| Light intensity | $77.90_1$ | <0.001 | <0.001 | $1.98_1$ | 0.160 | 0.389 | $335.34_1$ | <0.001 | *<0.001* |

(*Continued.*)

**Table 2.** (*Continued.*)

| predictors | $\chi^2_{d.f.}$ | p | adj. p | $\chi^2_{d.f.}$ | p | adj. p | $\chi^2_{d.f.}$ | p | adj p |
|---|---|---|---|---|---|---|---|---|---|
| Line × Density | $0.45_1$ | 0.503 | 0.737 | $0.05_1$ | 0.815 | 0.922 | $0.20_1$ | 0.654 | 0.847 |
| Line × Light intensity | $11.03_1$ | 0.001 | *0.005* | $<0.01_1$ | 0.983 | 0.984 | $0.71_1$ | 0.401 | 0.674 |

| predictors | Nematoda[§] | | | Hydrachnidia[§] | | | Planorbidae[†] | | |
|---|---|---|---|---|---|---|---|---|---|
| | $\chi^2_{d.f.}$ | p | adj. p | $\chi^2_{d.f.}$ | p | adj. p | $\chi^2_{d.f.}$ | p | adj p |
| intercept | — | — | — | — | — | — | — | — | — |
| Line | $12.83_1$ | <0.001 | *0.002* | $3.65_1$ | 0.056 | 0.170 | $0.09_1$ | 0.768 | 0.889 |
| Density | $0.25_1$ | 0.619 | 0.826 | $2.29_1$ | 0.131 | 0.353 | $0.62_1$ | 0.432 | 0.689 |
| Light intensity | $0.28_1$ | 0.594 | 0.812 | $0.04_1$ | 0.835 | 0.926 | $1.14_1$ | 0.286 | 0.569 |
| Line × Density | $0.70_1$ | 0.404 | 0.674 | $0.10_1$ | 0.755 | 0.889 | $2.11_1$ | 0.146 | 0.372 |
| Line × Light intensity | $0.23_1$ | 0.632 | 0.830 | $1.71_1$ | 0.191 | 0.443 | $0.97_1$ | 0.325 | 0.615 |

| predictors | Corbiculidae[†] | | | Ephemeroptera[§] | | | Simpson's diversity[#] | | |
|---|---|---|---|---|---|---|---|---|---|
| | $\chi^2_{d.f.}$ | p | adj p | $\chi^2_{d.f.}$ | p | adj p | $\chi^2_{d.f.}$ | p | adj p |
| intercept | — | — | — | — | — | — | — | — | — |
| Line | $0.82_1$ | 0.364 | 0.647 | $0.14_1$ | 0.706 | 0.875 | $0.23_1$ | 0.631 | 0.830 |
| Density | $0.35_1$ | 0.556 | 0.777 | $0.10_1$ | 0.753 | 0.889 | $0.54_1$ | 0.464 | 0.710 |
| Light intensity | $0.10_1$ | 0.754 | 0.889 | $1.96_1$ | 0.161 | 0.389 | $29.53_1$ | <0.001 | *<0.001* |
| Line × Density | $0.17_1$ | 0.679 | 0.867 | $0.01_1$ | 0.920 | 0.960 | $2.18_1$ | 0.193 | 0.362 |
| Line × Light intensity | $0.03_1$ | 0.872 | 0.945 | $0.17_1$ | 0.678 | 0.867 | $0.01_1$ | 0.913 | 0.960 |

(*Continued.*)

**Table 2.** (*Continued.*)

ecosystem metrics
LMMs with Gaussian distribution

| predictors | GPP $F_{d.f., d.f.res}$ | p | adj p | CR24 $F_{d.f., d.f.res}$ | p | adj p | benthic algae* $F_{d.f., d.f.res}$ | p | adj p |
|---|---|---|---|---|---|---|---|---|---|
| intercept | — | — | — | — | — | — | $45.17_{1,7.30}$ | <0.001 | 0.002 |
| Line | $0.74_{1,39.21}$ | 0.394 | 0.674 | $0.68_{1,38.99}$ | 0.416 | 0.674 | $0.54_{1,38.28}$ | 0.466 | 0.710 |
| Density | $6.12_{1,39.21}$ | 0.018 | 0.067 | $8.09_{1,39.15}$ | 0.007 | 0.030 | $5.63_{1,38.32}$ | 0.023 | 0.084 |
| Light intensity | $33.00_{1,38.02}$ | <0.001 | <0.001 | $36.74_{1,38.00}$ | <0.001 | <0.001 | $8.35_{1,38.11}$ | 0.006 | 0.027 |
| Line × Density | $0.04_{1,40.22}$ | 0.836 | 0.926 | $0.15_{1,40.08}$ | 0.702 | 0.875 | $0.43_{1,38.42}$ | 0.518 | 0.747 |
| Line × Light intensity | $<0.01_{1,39.20}$ | 0.984 | 0.984 | $0.01_{1,39.07}$ | 0.930 | 0.960 | $9.16_{1,38.21}$ | 0.004 | 0.022 |

| predictors | Pelagic algae* $F_{d.f., d.f.res}$ | p | adj p | $NH_4^+$ concentration* $F_{d.f., d.f.res}$ | p | adj p | SRP concentration* $F_{d.f., d.f.res}$ | p | adj p |
|---|---|---|---|---|---|---|---|---|---|
| intercept | — | — | — | — | — | — | — | — | — |
| Line | $1.37_{1,38.50}$ | 0.248 | 0.524 | $2.56_{1,38.03}$ | 0.118 | 0.324 | $0.10_{1,38.17}$ | 0.757 | 0.889 |
| Density | $8.99_{1,38.50}$ | 0.005 | 0.022 | $0.40_{1,38.03}$ | 0.531 | 0.754 | $0.01_{1,38.17}$ | 0.929 | 0.960 |
| Light intensity | $0.10_{1,38.00}$ | 0.749 | 0.889 | $27.62_{1,38.00}$ | <0.001 | <0.001 | $0.16_{1,38.00}$ | 0.610 | 0.819 |
| Line × Density | $0.60_{1,38.96}$ | 0.443 | 0.696 | $0.29_{1,38.06}$ | 0.595 | 0.812 | $<0.01_{1,38.33}$ | 0.957 | 0.969 |
| Line × Light intensity | $0.50_{1,38.49}$ | 0.484 | 0.728 | $1.23_{1,38.03}$ | 0.274 | 0.567 | $1.41_{1,38.16}$ | 0.242 | 0.524 |

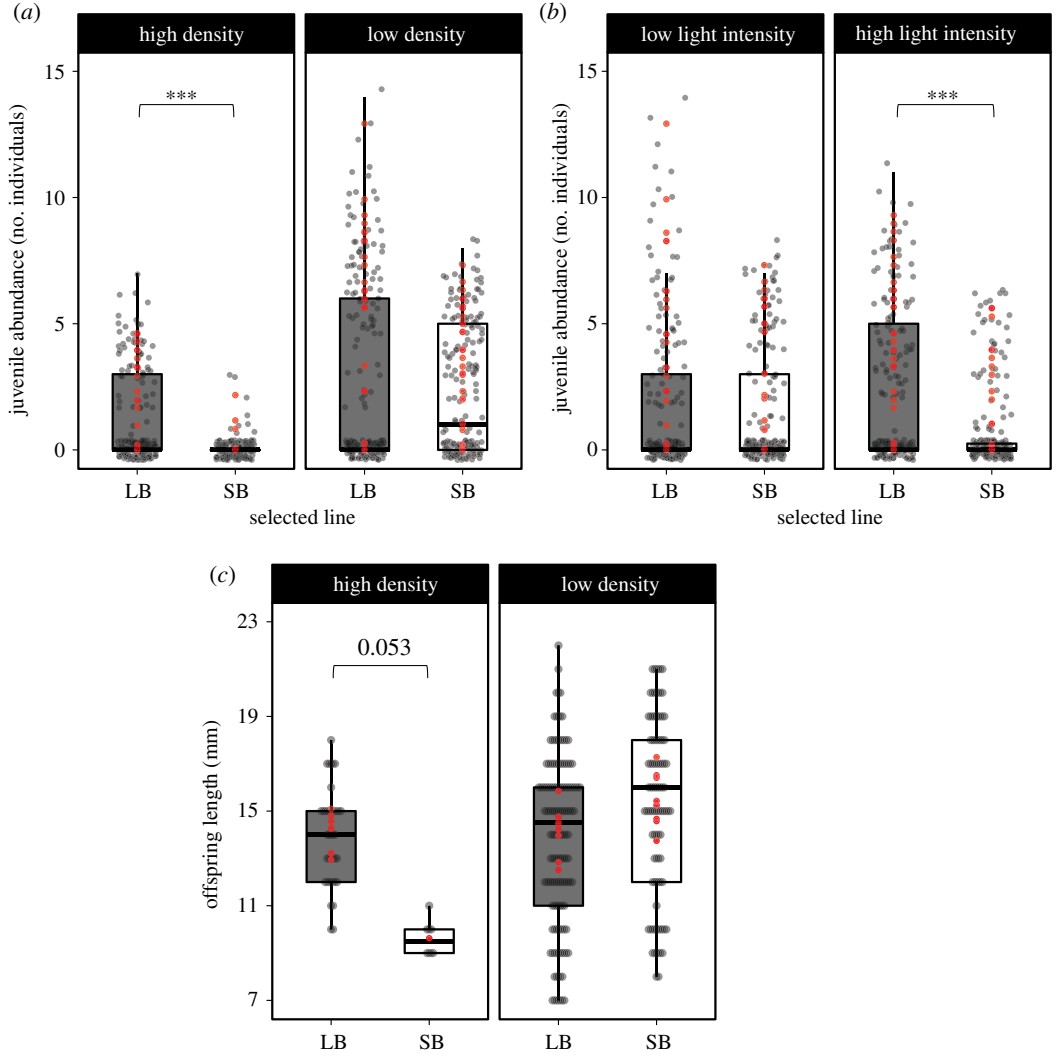

**Figure 2.** Boxplots (raw data; grey dots) showing the (*a*) density- and (*b*) light-dependent effects of size selection on juvenile abundance (number of individuals) and (*c*) the density-dependent effect of size selection on offspring length (mm). Red dots are predicted values (ŷ) from the models. $^{**}p < 0.01$, $^{***}p < 0.001$.

Effect sizes show that, overall, the ecological effects induced by size selection, density, light and fish introduction were of similar magnitude (*t* values ranged from −2.15 to 2.07; adj $p > 0.168$). Response to size selection led to the smallest effect on ecosystem metrics (figure 4). For community metrics, the effect of size selection was of similar magnitude to that induced by fish introduction, and of higher intensity than that induced by fish density variation (figure 4).

## 4. Discussion

By removing large-sized individuals, harvesting can induce rapid evolutionary change in consumer life history [3], but the consequences of such changes for the whole ecosystem remain empirically poorly explored [8]. Using medaka from previously size-selected lines (large-breeder LB and small-breeder SB) in a pond mesocosm experiment, we showed that small-breeder medaka had lower recruitment (i.e. juvenile abundance) than their large-breeder counterparts but only at high density or high light treatments. Contrary to our prediction, the somatic growth of recaptured fish did not differ between the two lines. This suggests that close to natural outdoor conditions alleviate the phenotypic response of medaka to size-dependent selection in the laboratory, as also observed in other experiments [43,44]. There was no evidence that population excretion differed between the two lines, but the abundances of Ostracoda and Nematoda were higher in SB than in LB populations. This

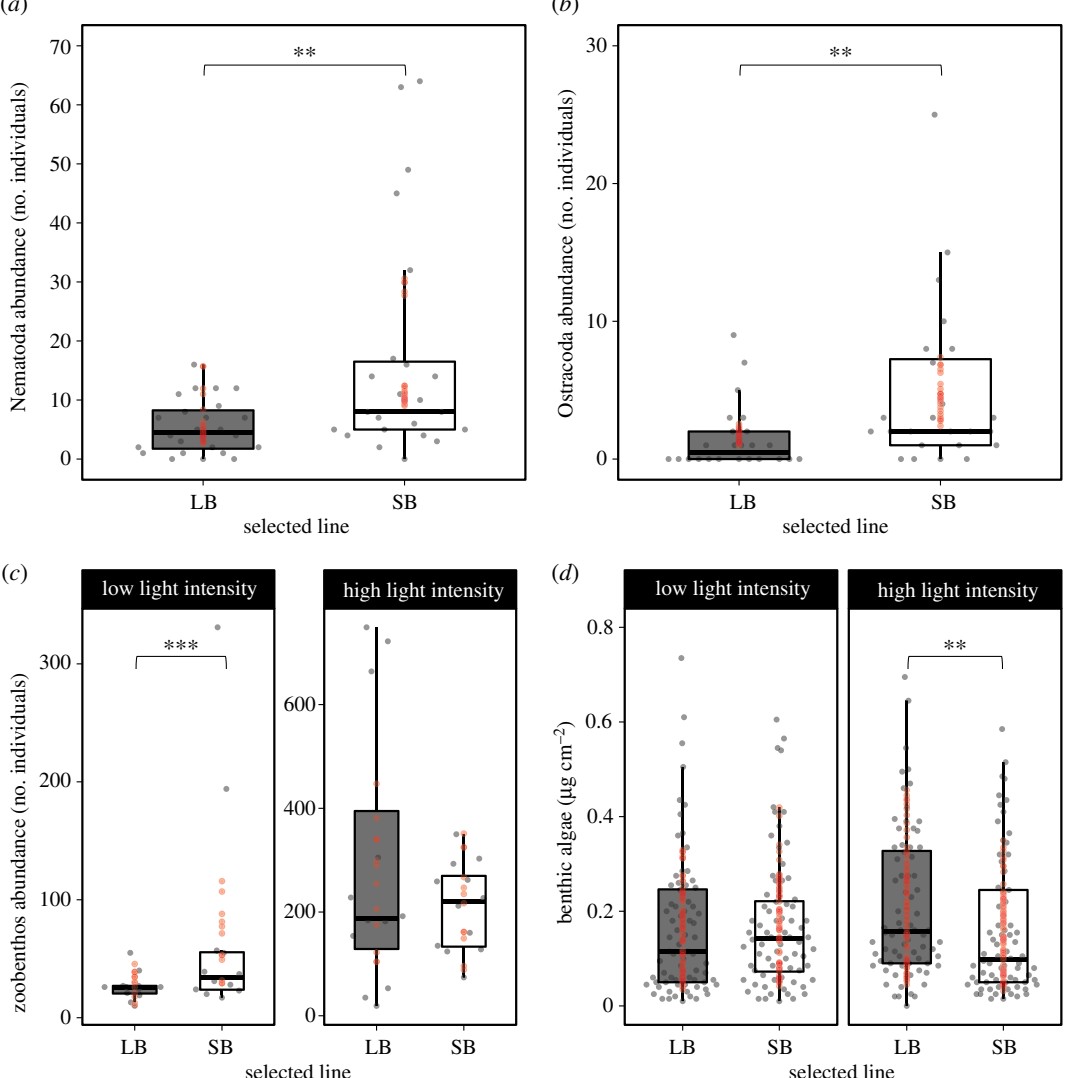

**Figure 3.** Boxplots (raw data; grey dots) showing the effects of size selection on (*a*) Nematoda and (*b*) Ostracoda abundances (number of individuals), and the light-dependent effects of size selection on (*c*) total zoobenthos abundance (number of individuals) and (*d*) benthic algae biomass (µg cm$^{-2}$). Red dots are predicted values (ŷ) from the models. $^{**}p < 0.01$, $^{***}p < 0.001$.

difference may have resulted in a stronger trophic cascade because benthic algae biomass increased with light intensity (bottom-up regulation) when grazers were controlled by LB medaka (top-down regulation). These outcomes indicate that life-history evolution due to adaptation to size-selective mortality can translate to prey community structure and ecosystem function, but that these effects strongly depend on environmental characteristics (Evolution × Environment).

Our findings highlight that increased fish density had a stronger negative impact on the juvenile abundance in the SB than in the LB populations, which might indicate that SB populations have a lower ability to cope with competition than LB medaka. Therefore, populations composed of fish selected for small size (harvest-like) will be more vulnerable to changes in density, which will impact on reference point for management [45,46]. In particular, fisheries surplus production models predict similar trajectories during decline and recovery [47], while harvest-induced evolution towards decreased ability to cope with increased population density predicts lower surplus production during recovery than decline. Larvae abundance was not significantly affected by the Line × Density interaction, suggesting no change in medaka fecundity. Despite the effect of the interaction Line × Density on offspring length was not significant with our conservative estimation of *p*-values (adj *p* = 0.053), we believe it is worth mentioning that in the high-density treatment, LB offspring seemed to grow on average 1.5 times faster than SB offspring, probably resulting in a shorter larvae–juvenile

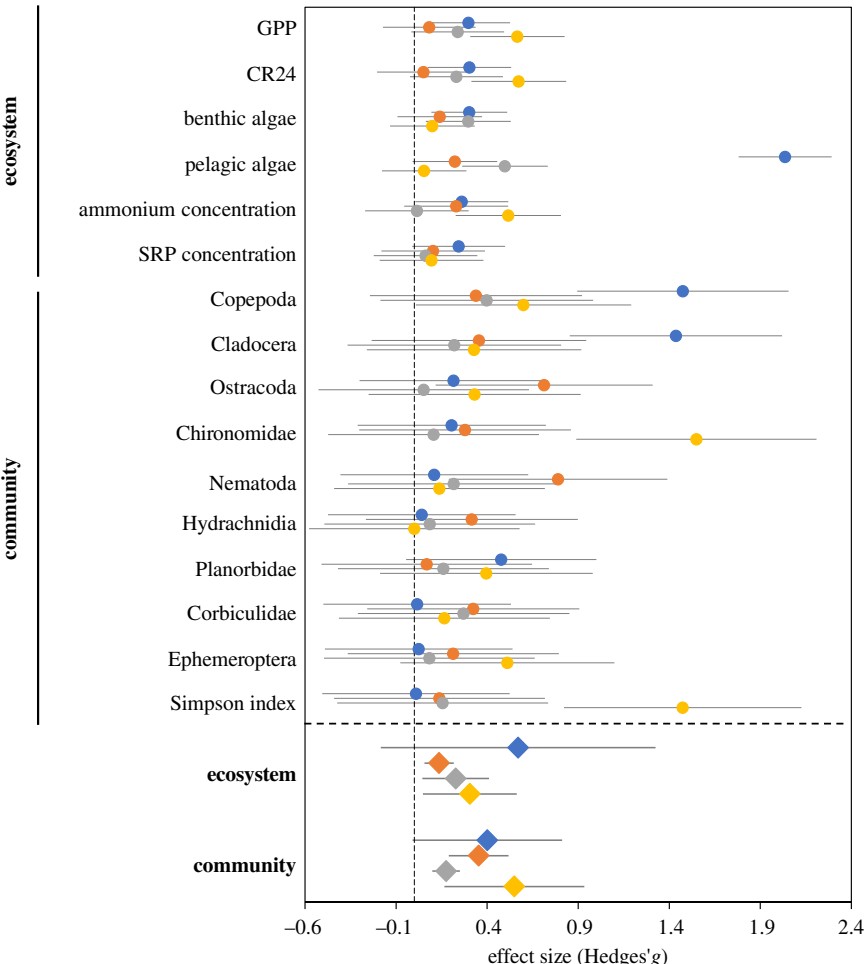

**Figure 4.** Effect size (Hedges' *g*) of fish introduction (blue), size selection (orange), fish density (grey) and light intensity (yellow) for each response variable (circle) and averaged ecosystem and community responses (triangle). Error bars represent 95% CI.

transition. Therefore, it seems reasonable to presume that density-dependent changes in juvenile abundance could be explained by two non-mutually exclusive mechanisms occurring during the larvae–juvenile transition, rather than changes in reproduction investment. First is an increased larvae–juvenile survival in LB populations compared with SB populations (in high fish density), second is an increased somatic grow rate of LB offspring that accelerates the larvae–juvenile transition and ultimately increases juvenile abundance. Altogether, these density-dependent effects highlight that LB fish cope better with competition, which could translate into higher population growth rate and maybe also a higher carrying capacity [23].

The Line × Light interaction on juvenile abundances closely paralleled the Line × Density interaction, suggesting similar mechanisms. Initially, our aim in varying light intensity was to vary primary production and test the prediction that SB medaka would benefit less from increased primary production than LB medaka. However, the shade nets also strongly reduced water evaporation, and the high-light treatment resulted in lower water volume and increased fish density per litre. This is perhaps why the light and density treatments have similar effects on juvenile medaka abundances, indicating that the density-mediated effects of light overwhelmed its primary productivity-mediated effects. It is worth noting, however, that in our pond experiment light had a positive effect on the whole pond metabolism and on the abundance of zoobenthos (i.e. total abundance, Ostracoda and Chironomidae abundances; electronic supplementary material, table S2.B), indicating that the primary productivity-mediated effect was indeed present. Therefore, the Line × Light interaction on juvenile abundance also suggests that SB populations did not benefit from increased productivity (high light intensity treatment), perhaps because increased light might benefit more to more efficient consumers such as LB medaka [27].

According to our predictions, we showed that the abundances of some of the benthic prey (i.e. Ostracoda and Nematoda) were higher in SB than in LB populations, demonstrating higher predation

rates of LB medaka. This suggests that SB medaka have reduced willingness to forage on benthos, as seen in earlier studies (silverside *Menidia menidia*: [28], medaka: [27]). We speculate that LB fish are more efficient foragers within the bottom substrate, where small and cryptic prey such as Nematoda and Ostracoda are difficult to find. Accordingly, fish selected for small size (i.e. SB medaka) may have a narrower diet breadth, thereby increasing their vulnerability to prey fluctuations and/or increased competition in natural environments. These findings echo and expand on studies showing that body size often correlates positively with diet diversity [48,49]. Bassar *et al.* [50] found that guppies from low-predation environments, where guppies form high population densities, consume more periphyton than guppies from high-predation environments, where guppy occur at low densities. This diet divergence in guppies is supposed to have evolved in response to competition selection at high population density, where preferred animal resources are scarce and guppies should thus adapt to also exploit primary producers. Our present results in medaka show that diet breadth may respond to size-dependent selection under constant population density and, hence, suggest that competition selection may possibly operate through size-dependent selection. Accordingly, it was recently shown in medaka that increased competition at high population density selects for larger body sizes [6].

The response of benthic algae biomass to the Line × Light intensity interaction is similar to that of juvenile abundance, perhaps indicating that interactive effects of the evolutionary response to size-selective mortality and the environment on fish biomass production can propagate into the ecosystem. Indeed, the low abundance of SB juveniles under high-light intensity probably reduced predation pressure on grazers, ultimately resulting in low benthic algae biomass due to the trophic cascade. By contrast, when predation is stronger due to high juvenile abundance—LB population under high-light intensity—we found that benthic algae biomass was also higher. However, benthic algae biomass did not change between the two lines under low-light intensity, even though LB medaka had an overall stronger effect than SB medaka in the benthic compartment. Although this should be treated with caution because the effect of light was confounded with a change in fish density due to evaporation, a plausible explanation could be that light increased benthic algae biomass (bottom-up regulation) only when grazers were controlled by LB medaka (top-down regulation). This highlights the interdependence between forces regulating ecosystem function, and further investigations are needed to address the potential role of consumer-driven nutrient recycling in mediating density- and/or light-dependent variations in algae biomass.

The effects of medaka evolution (i.e. intraspecific variability) on community responses were of similar intensity to those induced by the addition of the species, and of higher intensity than those induced by varying density. This confirms that intraspecific trait variation is a key driver of resource dynamics [51], and further demonstrates that rapid evolution caused by humans may substantially contribute to mediating intraspecific biodiversity–community structure relationships. Contrary to recent meta-analyses [10,11], we found that intraspecific variation was a weak predictor of ecosystem metrics. In the present study, nutrient excretion rates were similar between the two lines, perhaps because a 5% evolutionary difference in body size was not large enough to influence consumer-induced nutrient recycling. This could ultimately limit the magnitude of the effects of intraspecific variation on ecosystem functioning [11] and suggests that these effects are rather top-down than bottom-up mediated.

The present study shows that trait changes induced by size-dependent mortality are not limited to fish biomass production, but can scale up to ecosystems, thus supporting the existence of evolution-induced iBEF relationships [13]. We empirically demonstrated that populations composed of fish selected for smaller size were vulnerable to the increased density and also had a restricted foraging niche and/or lower willingness to forage. This could perhaps limit population recovery following fishing relaxation. Ultimately, the evolutionary response to size-selective mortality expands beyond populations dynamics [52,53], highlighting that accounting for evolution is crucial to implement ecosystem-based approaches to fisheries management.

Ethics. The experiment was approved by the Darwin Ethical committee (case file no. Ce5/2010/041) from the French Ministry of Education, Higher Education and Research.

Data accessibility. Data and codes that support the findings of this study are hosted in the Figshare Repository: https://figshare.com/s/fa17adc6acd0c149cd28.

Authors' contribution. C.E. and E.E. conceived the study and designed the experiment; C.E. coordinated the study; C.E., J.D. and J.S. conducted the samplings with the help of A.H., J.M., L.A.V. and B.D.P.; J.S. collected the count data; C.E. analysed the data and wrote the initial draft of the manuscript; E.E., B.D.P. and L.A.V. contributed to revisions. All authors approved the final version of the manuscript.

Competing interests. We declare we have no competing interests.

Funding. This work was supported by The Research Council of Norway (project nos. 251307/F20 and 255601/E40) and its mobility programme (project nos. 272354 to C.E. and 268218/MO to B.D.P.). E.E. was supported by IDEX SUPER (project Convergences no. J14U257 MADREPOP) and by Rennes Métropole (AIS 18C0356). The experiments realized in the CEREEP Ecotron Ile-De-France benefited from the support received by an 'Investissements d'Avenir' programme from the Agence Nationale de la Recherche (ANR-10-EQPX-13-01 Planaqua and ANR-11-INBS-0001 AnaEE France).

Acknowledgements. We are grateful to Clémentine Renneville and Arnaud Le Rouzic for initiating the medaka lines; David Carmignac and Romain Péronnet for their help maintaining the fish; and the platform PLANAQUA and the CEREEP Ecotron Ile-De-France for the access to the experimental facilities. We thank one anonymous reviewer and the Associate Editor for constructive inputs that improved the manuscript.

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
