## [Peer Review File · Royal Society Open Science]

Review History

RSOS-210842.R0 (Original submission)

Review form: Reviewer 1

Is the manuscript scientifically sound in its present form?

Yes

Are the interpretations and conclusions justified by the results?

Yes

Is the language acceptable?

Yes

Do you have any ethical concerns with this paper?

No

Have you any concerns about statistical analyses in this paper?

No

Recommendation?

Accept with minor revision (please list in comments)

Comments to the Author(s)

This manuscript presents an interesting study of the community and ecosystem-level effects of large and small size-selected medaka in high and low density and high and low light conditions. The key result is that the ecological effects of differentially size-selected lines were similar in magnitude to the effects of fish presence, fish density, and light intensity. Additionally, there were significant evolution-by-environment interactions in which the effects of small and large breeder lines differed but only under high density or high light conditions. Overall, the manuscript was a pleasure to read. I have a few small comments detailed below.

This study is an important contribution to the field, but it is missing reference to the other example that I know of testing the context-dependency of evolution-induced indirect effects (Environment X Evolution): Bassar et al. (2010) examines the effect of HP and LP guppies at high and low density on ecosystem variables such as algae standing stocks & invert biomass.

I appreciate how the hypotheses are clearly spelled out. This helps set up expectations for some of the more complicated interactive effects. Furthermore, I believe the hypotheses are tested with statistically sound methods.

Not being familiar with the system, it would be nice to have more details of the evolutionary differences between the two lines (ex: line 431) in the introduction or methods. Also, is there any estimate of heritability for growth rates and/or size? The number of generations of selection alone are not proof that differences between these lines are genetic (line 129).

Line 117: Is this describing a per capita increase?

Line 164: Please clarify what is meant by these blocks. Currently, "two treatments were doubled" is unclear. I interpret this as 2 treatments x 2 x 5 blocks which is only 30 tanks.

Line 168: While it is useful to know why light was used instead of a nutrient addition, saying this will hold nutrient levels constant seems unlikely (i.e. increasing primary production via more light would also increase nutrient uptake).

Line 176: What density were the zooplankton added, was it ambient or concentrated?

Line 179: Were the mesocosms sampled before fish were added? This would allow you to separate some of the effects of light intensity (aka strength of bottom-up effects).

Line 208: This should be reflected in n in table 1.

Line 313: Clarify if these are LB offspring in the low-density treatment.

Line: 315: The lack of effect of size selection on somatic growth is very surprising given the experimental design assumes the fish are different in this trait. I think it deserves more attention.

Line 357, 407-422. Because the light treatment was confounded with a change in density due to evaporation, I would suggest de-emphasizing conclusions about the relative strengths of bottom-up versus top-down.

Line 402: This foraging result could also be compared to the finding of Bassar et al. 2012 that low predation guppies include more periphyton in their diet.

Figure 1. I don't find this experimental design figure particularly useful.

Figure 4 is beautiful. What a nice summary of all the results. It makes the comparison between the magnitude of evolutionary and ecological treatments really easy.

Bassar, R.D., Marshall, M.C., Lopez-Sepulcre, A., Zandonà, E., Auer, S.K., Travis, J., Pringle, C.M., Flecker, A.S., Thomas, S.A., Fraser, D.F., Reznick, D.N., 2010. Local adaptation in Trinidadian guppies alters ecosystem processes. *Proc. Natl. Acad. Sci. U.S.A.* 107, 3616–3621

Bassar, R.D., Ferriere, R., Lopez-Sepulcre, A., Marshall, M.C., Travis, J., Pringle, C.M., Reznick, D.N., 2012. Direct and indirect ecosystem effects of evolutionary adaptation in the Trinidadian guppy (*Poecilia reticulata*). *Am. Nat.* 180, 167–185.

Decision letter (RSOS-210842.R0)

Dear Dr Evangelista

On behalf of the Editors, we are pleased to inform you that your Manuscript RSOS-210842 "Ecological ramifications of adaptation to size-selective mortality" has been accepted for publication in Royal Society Open Science subject to minor revision in accordance with the referees' reports. Please find the referees' comments along with any feedback from the Editors below my signature.

Please submit your revised manuscript and required files (see below) no later than 7 days from today's (ie 24-Aug-2021) date. Note: the ScholarOne system will 'lock' if submission of the revision is attempted 7 or more days after the deadline. If you do not think you will be able to meet this deadline please contact the editorial office immediately.

Kind regards,
Royal Society Open Science Editorial Office
Royal Society Open Science

on behalf of Professor Enrico Bertuzzo (Associate Editor) and Pete Smith (Subject Editor)
openscience@royalsociety.org

Associate Editor Comments to Author (Professor Enrico Bertuzzo):

Associate Editor: 1

Comments to the Author:

It was not possible to secure the report from one referee that initially agreed to review the manuscript. Instead of trying to seek an additional reviewer at this stage, which would further delay the editorial process, I decided to review the manuscript myself and add my comments to those provided by the first reviewer.

I am pleased to report that, in agreement with the other reviewer, I found the experiment well-designed and the manuscript relevant, well-written and easy to follow. In addition to the suggestions made by the reviewer, I recommend considering also the following comments in your revision:

Line 34: remove comma after faster.

Line 144: "F11" the numbering of the fish generation is not clear at this point (it was not previously introduced)- Please details better this part

Line 149: The journal has a wide audience which can be unfamiliar with the kinship coefficient. Please detail the calculation or provide a suitable reference.

Lines 341-348. This part is redundant and could be greatly summarized or removed.

Figure 1. Differently from the referee, I would suggest to keep this figure.

Calculation of GPP and Respiration. In the calculation shown, it seems to me that the authors are neglecting that the duration of the period t_2-t_1 is different from that of t_1-t_0 . Moreover, the difference changes during the experiment. Specifically, the respiration measured at night cannot be simply added to the NPP measured during the day to get the GPP, because the amount respired during the day could be different because of the different durations of the two periods. One should first calculate a rate (production/consumption per unit of time) or alternatively account for the different durations. The reference provided by the authors (Harmon et al. 2009) seems to perform the same calculations that the authors did. However, Harmon et al. 2009 in turn cites Downing, 2005 which indeed compute hourly rate of consumption/production of oxygen. Please clarify this point

Reviewer comments to Author:

Reviewer: 1

Comments to the Author(s)

This manuscript presents an interesting study of the community and ecosystem-level effects of large and small size-selected medaka in high and low density and high and low light conditions. The key result is that the ecological effects of differentially size-selected lines were similar in magnitude to the effects of fish presence, fish density, and light intensity. Additionally, there were significant evolution-by-environment interactions in which the effects of small and large breeder lines differed but only under high density or high light conditions. Overall, the manuscript was a pleasure to read. I have a few small comments detailed below.

This study is an important contribution to the field, but it is missing reference to the other example that I know of testing the context-dependency of evolution-induced indirect effects (Environment X Evolution): Bassar et al. (2010) examines the effect of HP and LP guppies at high and low density on ecosystem variables such as algae standing stocks & invert biomass.

I appreciate how the hypotheses are clearly spelled out. This helps set up expectations for some of the more complicated interactive effects. Furthermore, I believe the hypotheses are tested with statistically sound methods.

Not being familiar with the system, it would be nice to have more details of the evolutionary differences between the two lines (ex: line 431) in the introduction or methods. Also, is there any estimate of heritability for growth rates and/or size? The number of generations of selection alone are not proof that differences between these lines are genetic (line 129).

Line 117: Is this describing a per capita increase?

Line 164: Please clarify what is meant by these blocks. Currently, "two treatments were doubled" is unclear. I interpret this as 2 treatments x 2 x 5 blocks which is only 30 tanks.

Line 168: While it is useful to know why light was used instead of a nutrient addition, saying this will hold nutrient levels constant seems unlikely (i.e. increasing primary production via more light would also increase nutrient uptake).

Line 176: What density were the zooplankton added, was it ambient or concentrated?

Line 179: Were the mesocosms sampled before fish were added? This would allow you to separate some of the effects of light intensity (aka strength of bottom-up effects).

Line 208: This should be reflected in n in table 1.

Line 313: Clarify if these are LB offspring in the low-density treatment.

Line: 315: The lack of effect of size selection on somatic growth is very surprising given the experimental design assumes the fish are different in this trait. I think it deserves more attention.

Line 357, 407-422. Because the light treatment was confounded with a change in density due to evaporation, I would suggest de-emphasizing conclusions about the relative strengths of bottom-up versus top-down.

Line 402: This foraging result could also be compared to the finding of Bassar et al. 2012 that low predation guppies include more periphyton in their diet.

Figure 1. I don't find this experimental design figure particularly useful.

Figure 4 is beautiful. What a nice summary of all the results. It makes the comparison between the magnitude of evolutionary and ecological treatments really easy.

Bassar, R.D., Marshall, M.C., Lopez-Sepulcre, A., Zandonà, E., Auer, S.K., Travis, J., Pringle, C.M., Flecker, A.S., Thomas, S.A., Fraser, D.F., Reznick, D.N., 2010. Local adaptation in Trinidadian guppies alters ecosystem processes. *Proc. Natl. Acad. Sci. U.S.A.* 107, 3616–3621

Bassar, R.D., Ferriere, R., Lopez-Sepulcre, A., Marshall, M.C., Travis, J., Pringle, C.M., Reznick, D.N., 2012. Direct and indirect ecosystem effects of evolutionary adaptation in the Trinidadian guppy (*Poecilia reticulata*). *Am. Nat.* 180, 167–185.

===PREPARING YOUR MANUSCRIPT===

===PREPARING YOUR REVISION IN SCHOLARONE===

<https://royalsociety.org/journals/authors/author-guidelines/#supplementary-material> to include a suitable title and informative caption. An example of appropriate titling and captioning may be found at https://figshare.com/articles/Table_S2_from_Is_there_a_trade-off_between_peak_performance_and_performance_breadth_across_temperatures_for_aerobic_scops_in_teleost_fishes_/3843624.

Author's Response to Decision Letter for (RSOS-210842.R0)

See Appendix A.

Decision letter (RSOS-210842.R1)

Dear Dr Evangelista,

I am pleased to inform you that your manuscript entitled "Ecological ramifications of adaptation to size-selective mortality" is now accepted for publication in Royal Society Open Science.

on behalf of Professor Enrico Bertuzzo (Associate Editor) and Pete Smith (Subject Editor)
openscience@royalsociety.org

Appendix A

Point-by-Point response to Reviewers comments (RSOS-210842)

Associate Editor Comments to Author (Professor Enrico Bertuzzo):

Associate Editor #1: Comments to the Author:

Comment 1: It was not possible to secure the report from one referee that initially agreed to review the manuscript. Instead of trying to seek an additional reviewer at this stage, which would further delay the editorial process, I decided to review the manuscript myself and add my comments to those provided by the first reviewer.

I am pleased to report that, in agreement with the other reviewer, I found the experiment well-designed and the manuscript relevant, well-written and easy to follow. In addition to the suggestions made by the reviewer, I recommend considering also the following comments in your revision:

***** Our response: We are grateful to the Associate Editor and the Reviewer for their supporting feedback and constructive suggestions that helped improving the quality and clarity of the manuscript. We have carefully revised our manuscript (see detailed below), addressing both major and minor comments.**

Comment 2: Line 34: remove comma after faster.

***** Our response: Done (l. 34)**

Comment 3: Line 144: "F11" the numbering of the fish generation is not clear at this point (it was not previously introduced)- Please details better this part

***** Our response: The sentence has been rephrased as: "fish from the 11th generation (dubbed F11)" (l. 146).**

Comment 4: Line 149: The journal has a wide audience which can be unfamiliar with the kinship coefficient. Please detail the calculation or provide a suitable reference.

***** Our response: Further details about the kinship coefficient can be found in Le Rouzic et al. 2020, which is now clearly stated (l. 152-153).**

Comment 5: Lines 341-348. This part is redundant and could be greatly summarized or removed.

***** Our response: In order to avoid redundancy, we have decided to only present the main results of our experiment and to remove the hypotheses (l. 341-348).**

Comment 6: Figure 1. Differently from the referee, I would suggest to keep this figure.

***** Our response: We have decided to keep the Figure 1.**

Comment 7: Calculation of GPP and Respiration. In the calculation shown, it seems to me that the authors are neglecting that the duration of the period t_2-t_1 is different from that of t_1-t_0 . Moreover, the difference changes during the experiment. Specifically, the respiration measured at night cannot be simply added to the NPP measured during the day to get the GPP, because the amount respired during the day could be different because of the different durations of the two periods. One should first calculate a rate (production/consumption per

unit of time) or alternatively account for the different durations. The reference provided by the authors (Harmon et al. 2009) seems to perform the same calculations that the authors did. However, Harmon et al. 2009 in turn cites Downing, 2005 which indeed compute hourly rate of consumption/production of oxygen. Please clarify this point

***** Our response: We agree that clarifications and corrections were needed regarding ecosystem metabolism measurements. Daily community respiration (CR24; mg O₂ L⁻¹ day⁻¹) and GPP (mg O₂ L⁻¹ day⁻¹) were quantified using changes in dissolved oxygen (DO) measured at sunrise (t₀), sunset (t₁) and the following sunrise (t₂) (i.e. over a 24h period). Community respiration at night (CR_{night}; mg O₂ L⁻¹) was calculated as follows:**

$$CR_{\text{night}} = DO_{t_1} - DO_{t_2}$$

Total daily respiration (CR24; mg O₂ L⁻¹ day⁻¹) was calculated by multiplying the average hourly nighttime respiration rate by 24 (Bott, 2006):

$$CR24 = CR_{\text{night}} / hr_{\text{night}} \times 24$$

where hr_{night} is the number of night hours for each sampling date

Daily gross primary productivity (GPP; mg O₂ L⁻¹ day⁻¹) was calculated as follows:

$$GPP = CR24 + (DO_{t_1} - DO_{t_0})$$

These clarifications have been added to the revised version of the manuscript (Supplementary S1: l.44 - 52) and “respiration” has been replaced by “daily community respiration” or “CR24” to avoid confusion (Supplementary: l.17, l.119, l.223, l.231; Main document: l.197, l.202, l.275). The reference of Bott (2006) has been added to the reference list (Supplementary: l. 240 - 241).

In addition, models were re-run with these new values of CR24 and GPP, but results did not change (Table 2, Table S2.A-B, Table S4.A). Figures were also updated (Fig.4 and Fig. S2.A,C).

Bott, T. L. (2006). Primary productivity and community respiration. In: Methods in Stream Ecology, 2nd Edition. Elsevier, New York, pp 263–290.

Reviewer comments to Author:

Reviewer #1: Comments to the Author(s)

Comment 1: This manuscript presents an interesting study of the community and ecosystem-level effects of large and small size-selected medaka in high and low density and high and low light conditions. The key result is that the ecological effects of differentially size-selected lines were similar in magnitude to the effects of fish presence, fish density, and light intensity. Additionally, there were significant evolution-by-environment interactions in which the effects of small and large breeder lines differed but only under high density or high light conditions. Overall, the manuscript was a pleasure to read. I have a few small comments detailed below.

This study is an important contribution to the field, but it is missing reference to the other example that I know of testing the context-dependency of evolution-induced indirect effects (Environment X Evolution): Bassar et al. (2010) examines the effect of HP and LP guppies at high and low density on ecosystem variables such as algae standing stocks & invert biomass.

I appreciate how the hypotheses are clearly spelled out. This helps set up expectations for some of the more complicated interactive effects. Furthermore, I believe the hypotheses are tested with statistically sound methods.

***** Our response: We are grateful to the Reviewer for providing positive and constructive feedback. We acknowledge that the previous version of the manuscript was missing relevant literature, but this has been corrected in the revised version and the article of Bassar et al. (2010) has been added (l. 88, 477 - 480).**

Comment 2: Not being familiar with the system, it would be nice to have more details of the evolutionary differences between the two lines (ex: line 431) in the introduction or methods. Also, is there any estimate of heritability for growth rates and/or size?

***** Our response: We added the following sentence in the Methods: "On average at 75dph, SL was 20.7 mm in small breeders and 22.0 mm in large breeders (a 5.7% difference), and probability of being mature was 91.7% in Small breeders and 77% in Large breeders (a 18.0% difference) (Renneville et al. 2020)" (l. 143 - 145). Realized heritability (h^2) for body size was 0.059 (\pm 0.023 SE) and -0.027 (\pm 0.014 SE) for the Large- and Small-breeder lines, respectively (Le Rouzic et al. 2020). Although these information might be interesting to some readers, we believe they are not crucial for the general aims and conclusions of the present manuscript. We thus preferred keeping the manuscript simple and not to report these values, but we can do so if the Editor prefers.**

Comment 4: Line 164: Please clarify what is meant by these blocks. Currently, "two treatments were doubled" is unclear. I interpret this as 2 treatments x 2 x 5 blocks which is only 30 tanks.

***** Our response: The sentence has been rephrased as: "Due to space constraints, treatments were arranged in 5 blocks (i.e. 12 mesocosms per block), within which two treatments were replicated twice" (l. 167 - 169).**

Comment 5: Line 168: While it is useful to know why light was used instead of a nutrient addition, saying this will hold nutrient levels constant seems unlikely (i.e. increasing primary production via more light would also increase nutrient uptake).

***** Our response: True, and the sentence has been rephrased by removing "while holding nutrient constant" (l. 171 - 172).**

Comment 6: Line 176: What density were the zooplankton added, was it ambient or concentrated?

***** Our response: We used a concentrate density of zooplankton and this is now clarified (l. 179).**

Comment 7: Line 179: Were the mesocosms sampled before fish were added? This would allow you to separate some of the effects of light intensity (aka strength of bottom-up effects).

***** Our response: Unfortunately, we did not sample our mesocosms before fish introduction.**

Comment 8: Line 208: This should be reflected in n in table 1.

***** Our response: True, and this mistake has been corrected (Table 1).**

Comment 9: Line 313: Clarify if these are LB offspring in the low-density treatment.

***** Our response: We have corrected the sentence: "LB offspring seemed to grow faster than SB ones in the high-density treatment." (l. 314 - 315).**

Comment 10: Line: 315: The lack of effect of size selection on somatic growth is very surprising given the experimental design assumes the fish are different in this trait. I think it deserves more attention.

***** Our response: Medaka from the two lines exhibited different growth trajectory under laboratory conditions (Evangelista et al. 2020, Renneville et al. 2020), but no difference was detected here. This suggests that close to natural outdoor conditions alleviate the phenotypic response of fish to size-dependent selection in the laboratory, as also observed in other experiments (Biro and Post 2008, Sutter et al. 2012). This is now mentioned in the revised manuscript (l. 343 - 346) and the references have been added to the reference list (l. 493 - 495, 599 - 602).**

Biro PA, Post JR. 2008 Rapid depletion of genotypes with fast growth and bold personality traits from harvested fish populations. *Proceedings of the National Academy of Sciences* 105, 2919–2922.

Sutter DAH, Suski CD, Philipp DP, Klefoth T, Wahl DH, Kersten P, Cooke SJ, Arlinghaus R. 2012 Recreational fishing selectively captures individuals with the highest fitness potential. *Proceedings of the National Academy of Sciences* 109, 20960–20965.

Comment 11: Line 357, 407-422. Because the light treatment was confounded with a change in density due to evaporation, I would suggest de-emphasizing conclusions about the relative strengths of bottom-up versus top-down.

***** Our response: We agree with the reviewer and we have thus revised our manuscript accordingly. Specifically, we have removed "and are mostly mediated via top-down regulation" (l. 353), as well as "top-down" (l. 417). The paragraph has also been re-structured to better highlight that we cannot make strict conclusions because the effect of light also influences fish density (l. 416 - 424).**

Comment 12: Line 402: This foraging result could also be compared to the finding of Bassar et al. 2012 that low predation guppies include more periphyton in their diet.

***** Our response: In line with Reviewer's comment we added the following sentences in line 402: "Bassar et al. (2012) found that guppies from low-predation environments, where guppies form high population densities, consume more periphyton than guppies from high-predation environments, where guppy occur at low densities. This diet divergence in guppies is supposed to have evolved in response to competition selection at high population density, where preferred animal resources are scarce and guppies should thus adapt to also exploit primary producers. Our present results in medaka show that diet breadth may respond to size-dependent selection under constant population density and, hence, suggest that competition selection may possibly operate through size-dependent selection. Accordingly, it was recently shown in medaka that increased competition at high**

population density selects for larger body sizes (Bouffet-Halle et al. 2021)" (l. 399 - 408, l. 481 - 484).

Comment 13: Figure 1. I don't find this experimental design figure particularly useful.

***** Our response: As suggested by the Associate Editor, we decided to keep this Figure (see AE comment 6)**

Comment 14: Figure 4 is beautiful. What a nice summary of all the results. It makes the comparison between the magnitude of evolutionary and ecological treatments really easy.

***** Our response: We thank the Reviewer for her/his enthusiasm for Fig.4.**

Bassar, R.D., Marshall, M.C., Lopez-Sepulcre, A., Zandonà, E., Auer, S.K., Travis, J., Pringle, C.M., Flecker, A.S., Thomas, S.A., Fraser, D.F., Reznick, D.N., 2010. Local adaptation in Trinidadian guppies alters ecosystem processes. *Proc. Natl. Acad. Sci. U.S.A.* 107, 3616–3621

Bassar, R.D., Ferriere, R., Lopez-Sepulcre, A., Marshall, M.C., Travis, J., Pringle, C.M., Reznick, D.N., 2012. Direct and indirect ecosystem effects of evolutionary adaptation in the Trinidadian guppy (*Poecilia reticulata*). *Am. Nat.* 180, 167–185.